# Clinical Experience of Emergency Appendectomy under the COVID-19 Pandemic in a Single Institution in South Korea

**DOI:** 10.3390/medicina58060783

**Published:** 2022-06-09

**Authors:** Yun Suk Choi, Jin Wook Yi, Chris Tae Young Chung, Woo Young Shin, Sun Keun Choi, Yoon Seok Heo

**Affiliations:** Department of Surgery, College of Medicine, Inha University Hospital, Incheon 400-711, Korea; yunsukki@gmail.com (Y.S.C.); jinwook.yi@inha.ac.kr (J.W.Y.); tylight8@gmail.com (C.T.Y.C.); mesik@hanmail.net (W.Y.S.); gshur@inha.ac.kr (Y.S.H.)

**Keywords:** appendicitis, COVID-19, emergencies

## Abstract

*Background and Objectives*: The COVID-19 pandemic has brought serious changes in healthcare systems worldwide, some of which have affected patients who need emergency surgery. Acute appendicitis is the most common surgical disease requiring emergency surgery. This study was performed to determine how the COVID-19 pandemic has changed the treatment of patients with acute appendicitis in South Korea. *Materials and Methods*: We retrospectively reviewed a medical database that included patients who underwent surgery for acute appendicitis in our hospital from January 2019 to May 2021. We classified the patients into two groups according to whether they were treated before or after the COVID pandemic and 10 March 2020 was used as the cutoff date, which is when the World Health Organization declared the COVID pandemic. *Results*: A total of 444 patients were included in the “Pre-COVID-19” group and 393 patients were included in the “COVID-19” group. In the “COVID-19” group, the proportion of patients with severe morbidity was significantly lower. The time that the patients spent in the emergency room before surgery was significantly longer in the ”COVID-19” group (519.11 ± 486.57 min vs. 705.27 ± 512.59 min; *p*-value < 0.001). There was no difference observed in the severity of appendicitis or in the extent of surgery between the two groups. *Conclusions*: During the COVID-19 pandemic, a statistically significant time delay (186.16 min) was needed to confirm COVID-19 infection status. However, there was no clinical difference in the severity of appendicitis or in the extent of surgery. To ensure the safety of patients and medical staff, a COVID-19 PCR test should be performed.

## 1. Introduction

At the end of 2019, an atypical pneumonia caused by a novel human coronavirus (COVID-19) was first reported in Wuhan, China, and it subsequently spread all over the world. The World Health Organization (WHO) declared that the COVID-19 outbreak was a Public Health Emergency of International Concern on 30 January 2020 and announced the global pandemic on 11 March 2020 [1]. Over the next 2 years, the COVID-19 pandemic situation has changed the whole healthcare system, including the control of COVID-19 infection and general medical care, such as elective and emergency surgical procedures. In early 2020, when the number of COVID-19-infected patients had surged, medical resources were concentrated on the treatment of these unknown coronavirus diseases, resulting in the collapse of the medical system, which made it impossible to treat existing general medical patients and patients who needed emergency services.

Acute appendicitis is one of the most common surgical diseases that requires emergency surgery among the various abdominal organ diseases. The lifetime incidence of appendicitis is reported to be 1 in 15 people, with a prevalence of 8.6% in men and a prevalence of 6.7% in women according to statistics for the United States [2,3]. Although there have been several reports of treating appendicitis with medical treatment, the most common treatment for appendicitis is surgical appendectomy. If diagnosis or surgical treatment is delayed, perforation of the appendix can result, which is related to serious complications, such as septic shock and mortality [4,5]. Overall, mortality due to a perforated appendix was reported to be 0.2–0.8% in a previous study [6]. Given this, to reduce the complications associated with appendicitis, diagnosis and surgical treatment of appendicitis should be performed quickly and without delay. Several diagnostic modalities are available but laboratory inflammatory markers, various scoring systems, computed tomography (CT) scan and abdominal ultrasonography have been used most frequently [7,8].

During the current COVID-19 pandemic, the world’s surgical societies have recommended that delays in some cases of planned surgery be considered and it has also been recommended that emergency surgery cases be classified and performed in a certain order according to the severity of patients’ medical situations [9]. Therefore, there is a concern that patients’ clinical conditions may deteriorate when surgery is delayed, especially in patients who have acute appendicitis (which would have been treated immediately in the past but where treatment might be delayed for patients presenting during the COVID-19 pandemic). In other medical or surgical emergencies, such as myocardial infarction and testicular torsion, there have been some reports that the incidence of disease complications increased during the COVID-19 pandemic [10,11]. There were several reports that the incidence of complicated acute appendicitis and rates of non-surgical treatment have increased compared to pre-COVID-19 levels [12,13,14,15,16,17]. The purpose of our study was to determine whether there was any meaningful change in clinical course in appendicitis patients who were seen after the COVID-19 pandemic compared to those seen prior to the pandemic. We aimed to compare the results of patients who underwent acute appendicitis surgery at a tertiary medical institution in South Korea 1 year prior to and 1 year after the date on which the COVID-19 pandemic was announced.

## 2. Methods

### 2.1. Ethics

The ethics of this study were approved by the Institutional Review Board of the author’s institution (IRB number: 2021-11-008; approval date: 22 November 2021).

#### 2.1.1. Data Collection and Patient Grouping

We retrospectively reviewed the medical data of acute appendicitis patients who received an emergent appendectomy at our hospital from January 2019 to May 2021. Patients who underwent delayed appendectomy or medical management for various reasons were excluded. The study period was determined to be 1 year and was based on the COVID-19 pandemic announcement date (11 March 2020) by the WHO. A total of 837 patients received emergency appendectomy during the study period. The patients were divided into two groups according to surgery dates: the “Pre-COVID-19” group included the 444 patients who had an appendectomy before 10 March 2020; the “COVID-19” group included the 393 patients who had an appendectomy after 11 March 2020.

We collected clinical data, including the patients’ general characteristics (age, sex, body mass index (BMI, kg/m^2^) and American Society of Anesthesiology (ASA) score); the time spent in the emergency room (ER) before the surgery; the operation-related variables (operation time, surgical approach, extent of surgery and drain placement); and laboratory values (WBC (white blood cell), Hb (hemoglobin), ANC (absolute neutrophil count) and CRP (C-reactive protein)) on the day before the surgery and the second day after the surgery. We also collected the pathological results of the appendicitis patients after surgery and the factors that were associated with postoperative hospital readmissions within 30 days after surgery.

The primary outcome in this study was the rate of complicated appendicitis and the secondary outcomes were the length of hospital stay, the rate of extended surgery and the readmission rate.

#### 2.1.2. The Emergency Surgery Protocol under the COVID-19 Pandemic in Our Hospital

All patients who received an emergency appendectomy initially visited the emergency department in our hospital. The diagnosis of appendicitis was based on the patient’s history, physical examination and imaging examinations, including a computed tomography (CT) scan or abdominal ultrasound. All of the preoperative general condition evaluations, such as blood chemistries, chest X-ray, electrocardiograms, serology and blood type tests, were the same in both the “Pre-COVID-19” and “COVID-19” groups. However, after the date of the COVID-19 pandemic announcement, all surgery patients were required to have nasal and throat swab COVID-19 PCR tests before surgery. All patients were allowed to enter the operating room after their COVID-PCR test results were verified. Even when a COVID-19 PCR test was negative, emergency surgery was performed in an isolated surgery room and the surgical staff were asked to wear PPE (personal protective equipment) in order to be able to assist in surgery due to the possibility of false-negative COVID-19 test results. If a COVID-19 PCR test was positive, surgery had to be performed in the negative pressure-supported operating room and the surgical staff were required to wear level I infection protection suits.

### 2.2. Surgical Method

The surgical approach was defined according to the method used to the perform surgery—open or laparoscopic. The choice of surgical approach was decided by the individual operating surgeon. All surgeries were performed under emergency.

Open appendectomy: A 5–7 cm modified McBurney or Rockey–Davis incision was made depending on the surgeon’s decision. After opening the abdominal wall, the appendix was identified and a periappendiceal tissue dissection was performed. The appendiceal vessel was ligated with an absorbable suture. The appendix base was ligated using a double-tie method, transected and removed. The exposed mucosa was cauterized with an electrosurgical device, such as a Bovie device. Stump inversion was carried out by placing a purse-string suture.

Laparoscopic appendectomy: An infraumbilical port was inserted initially for the laparoscope. After CO_2_ was insufflated at a pressure of 12 mmHg, an additional 2 ports were inserted (5 mm, 2 sites, or 5 mm, 12 mm, depending on the surgeon’s preference). Under the laparoscope, the appendix was identified and the periappendiceal tissue dissection was performed. The mesoappendix and appendiceal vessel were dissected using an ultrasonic shear energy device (Harmonic, Ethicon Endo-surgery, Cincinnati, OH, USA). The appendiceal base was tied using endoloops (Vycril Endoloop-0, Ethicon Endo-surgery, Cincinnati, OH, USA) or polymetric clips (Hem-o-lock, Teleflex, Morrisville, NC, USA). The appendiceal mucosal tip was cauterized with an electrical device, such as a Bovie device.

Single-port laparoscopic appendectomy: A single port (Gloveport, Nelis, Bucheon, Gyeonggi-do, Korea) was used for laparoscopy instead of the traditional 3 ports. A 2–3 cm transumbicial incision was made and the single port was inserted for the laparoscope. Other surgical methods are consistent with the conventional laparoscopic appendectomy.

### 2.3. Definition of the Terms Used in This Study

“Time spent in ER before surgery” was defined as the elapsed time (minutes) from the patient’s ER admission to entering the operating room (OR). “Hospital day” was defined as from the day of admission to the emergency room to the day of discharge after surgery. “Severity of appendicitis” was classified based on the patient’s postoperative pathological results. Uncomplicated appendicitis was defined as the presence of a normal appendix, an appendix with mild inflammation, suppurative appendicitis or other conditions. Complicated appendicitis was defined as gangrenous and perforated appendicitis. “Extent of surgery” was divided into two categories based on the extent of surgery, regardless of whether an open or a laparoscopic approach was used: an appendectomy alone or extended surgery including a partial cecectomy, an ileocecectomy or a right hemicolectomy. Postoperative complications were defined as undesirable and unintended events that occurred in patients within 30 days after the date of surgery and that were clinically considered to be due to a direct association with the appendectomy.

### 2.4. Statistics

The statistical analysis was performed using SPSS version 24.0 (SPSS Inc., Chicago, IL, USA). The chi-square test or Fisher’s exact test was applied for the cross-table analysis according to the sample size. Unpaired *t*-tests were used to compare the means between the two groups. A *p*-value under 0.05 was considered statistically significant.

## 3. Results

Table 1 shows the clinical characteristics of the appendicitis patients in the two groups. No significant differences in the patients’ ages, sex, BMIs, or hospital stays were observed between the two groups, as listed in Table 1. ASA scores were significantly lower in the “COVID group” (2.09 ± 0.62 vs. 1.96 ± 0.48; *p* = 0.001). The proportion of patients with ASA scores of III (severe systemic disease) was significantly lower in the “COVID-19” group than in the “Pre-COVID-19” group (24.4% in “Pre-COVID-19” vs. 9.4 in the “COVID-19” group). Operation time was significantly shorter in the “COVID-19” group (58.37 ± 30.84 min vs. 70.62 ± 30.96 min; *p* < 0.001). Time spent in ER before surgery was significantly longer in the “COVID-19” group (519.11 ± 486.57 min vs. 705.27 ± 512.59 min; *p* < 0.001). Despite the requirement for routine COVID-19 PCR tests, there were no patients who were confirmed to have COVID-19 among the “COVID-19” patients.

There was no significant difference in the rate of complicated appendicitis, including gangrenous or perforated appendicitis, between the two groups (18.9% vs. 22.9%; *p* = 0.154), as described in Table 2. The pathological diagnoses were also not different between the two groups. Regarding the extent of surgery, the proportion of patients who had extended surgery, including partial cecectomy, ileocecectomy or right hemicolectomy, was not significantly different between the two groups (4.5% vs. 5.1%; *p* = 0.687). The surgical approach (open versus laparoscopic) was also not different between the two groups. The need for an intraoperative drain placement was more common in the “Pre-COVID-19” group. The rate of readmission due to surgical complications within 30 days of surgery was significantly higher in the “COVID-19” group, as listed in Table 2.

The laboratory test results are described in Table 3. There was no significant difference between the two groups regarding the laboratory results that were performed on the preoperative day. The WBC count and ANC, taken on the second postoperative day, were significantly higher in the “COVID-19” group. However, all of the values were within the normal range in both groups, as shown in Table 3.

## 4. Discussion

As of January 2022, there were no published studies on the time that appendicitis patients must spend waiting during the in-hospital period for the results of a COVID-19 screening test. Several studies have reported that, during the COVID-19 pandemic, there was a prolonged delay for appendicitis patients to undergo surgery during the hospitalization period [18,19]. These studies proposed that delayed surgery may have contributed to the development of severe complications in appendectomy patients [17,20]. However, these studies only reported the situation during the time of complete lockdowns and when countries had implemented stay-at-home strategies. In those contexts, a proper diagnosis and treatment could be profoundly delayed. Prior studies have also reported a small number of cases among the 100 appendectomy cases after the COVID-19 pandemic. In addition, COVID-19 quarantine status and medical resources were significantly different among the various countries, which may have also affected the increased severity of the appendectomy patients described in previous reports.

There was a large-volume meta-analysis published that reported that during the COVID-19 pandemic the rate of complicated appendicitis and non-operative management of appendicitis increased [21]. There have also been many other reports that the incidence of complicated appendicitis has increased compared to that before the COVID-19 pandemic [12,13,14,16,22,23,24,25,26,27,28,29]. The HRs (hazard ratios) compared to those of pre-COVID-19 pandemic patients were from 1.13 to 3.15 (average HR = 1.63, CI = 1.33–2.01; *p* < 0.001). Factors that may have caused the increase in complicated appendicitis include the increase in durations from symptom onset to time of admission due to social distancing or isolation and delays in the time from admission to surgery because of hospital pandemic protocols [22]. These factors may have differed across countries, times, etc. There was a study similar to ours conducted in Korea at a similar time [30]. It was reported that the rate of complicated appendicitis increased compared to pre-COVID-19 pandemic levels (82/161 (50.9%) vs. 201/330 (60.9%); *p* < 0.036). However, there were no differences in postoperative length of hospital stay and complication rates between the two groups. The readmission rate within 30 days was lower compared to the pre-COVID-19 pandemic rate (6/161 (3.7%) vs. 2/330 (0.7%); *p* = 0.017).

There are various criteria that can be used to define complicated appendicitis, such as preoperative imaging studies, pathological results and surgeon’s subjective opinions based on intraoperative findings (Gomes’ laparoscopic grading system) [31,32,33]. Imaging studies and surgeon opinion criteria could not be used due to insufficient data in this retrospective study. This study defined complicated appendicitis based on pathological results. According to our study, there was no significant difference in the clinical course of acute appendicitis patients who received an emergency appendectomy between the “Pre-COVID-19” and the “COVID-19” groups. There was no statistically significant difference in the rate of extended surgery beyond simple appendectomy, which was only about 5% between the “Pre-COVID-19” and the “COVID-19” groups. There were also no significant differences in the pathological diagnoses or surgical treatments between the two groups. In our study, the in-hospital waiting time from ER admission to surgery was only 186 min (approximately 3 h), which was the time required for the COVID-19 PCR result to become available. It was not considered sufficient to determine the pathological result and change the clinical course of patients with appendicitis according to our study. One meta-analysis reported that in 2018, a hospital delay of 24 h did not increase the risk of complicated appendicitis [34]. As such, we suggest that a preoperative COVID-19 PCR test for all appendicitis patients before surgery can improve the safety of both patients and medical staff without increasing the complication rate of appendicitis.

The readmission rate within 30 days due to postoperative surgical complications was higher in the “COVID-19” group. The readmission rate in our study was higher than that reported in another study [35]. We think that this higher rate of postoperative complications might be due to factors associated with the surgeons who performed the surgery. The authors’ hospital is a tertiary institution and a medical university-affiliated hospital. As such, acute appendicitis surgeries are performed by numerous general surgery residents under the supervision of surgeon staff for training purposes. During the “Pre-COVID-19” period, we had six surgical residents but there were only four residents who were working in the “COVID-19” period. While performing a small number of surgical tasks, these residents also shared some of the treatment of COVID-19 patients, so fatigue could have occurred. We think that this hard-pressed situation may have affected the increased number of complications that occurred during the “COVID-19” period. Fortunately, most of the postoperative complications reported in this study were not severe; these complications included mechanical ileus, postoperative pain, fluid collection, and there was a small number of complications in the patients (13 patients, 3.3%). There were no major complications, such as operation site leakage, perforation or peritonitis, that required significant treatment, such as reoperation. If there is an increase in the number of residents in the future and if the COVID-19 crisis abates, these complications are expected to decrease again. We think that additional research is needed regarding this aspect.

The time spent in ER before surgery was 3 h longer in the “COVID-19” group compared to the “Pre-COVID-19” group, as mentioned in Table 1 (519.11 ± 486.57 min vs. 705.27 ± 512.59 min). This was the amount of the time that was needed to obtain the COVID-19 PCR results from upper respiratory tract specimens. To increase the efficiency of analyzing the numerous COVID-19 specimens, our hospital is currently conducting and reporting COVID-19 tests six times a day. Fortunately, all of the COVID-19 tests in the “COVID-19” group were negative in our hospital and there was no hospital spread of the COVID-19 virus between the appendectomy patients and the medical staff. According to previous reports from other institutions, prolonged delays of more than 12 to 24 h for appendectomies can significantly increase complication rates. However, a 3 h surgical delay to determine the patient’s COVID-19 status before surgery is suitable for both patients and medical staff to prevent the in-hospital spread of COVID-19. Therefore, during the COVID-19 pandemic, we suggest that an appendectomy can be performed safely after the routine COVID-19 PCR testing of samples from the patient’s upper respiratory tract.

The ASA scores of patients in the “COVID-19” group were significantly lower than those in the “Pre-COVID-19” group, as described in Table 1. We think that this phenomenon was due to the so-called “social distance strategy” that has been implemented since the onset of the COVID-19 pandemic in South Korea. As such, patients who have a good general condition and do not need help to move from place to place can visit the hospital more easily when they feel abdominal pain. In contrast, for patients who have severe comorbidities and require other people’s help to visit the hospital, it is difficult to ask for help from people because of the required social distancing due to the pandemic. For this reason, if acute abdominal symptoms occurred in patients who were in ill health, it is thought that these patients may have been taken to the nearest hospital, which was not necessarily a tertiary university hospital (such as the author’s institution). Since this hypothesis is not clearly proven, further research is required to prove this phenomenon.

Regarding surgical approach, there was no difference in the open versus laparoscopic methods in our hospital, as mentioned in Table 1. Several studies and guidelines have described the safety concerns related to general anesthesia and laparoscopic surgery in patients with suspected viral infections due to the risk of transmission of the virus through the air or through smoke [36,37]. However, in our hospital, all patients had a COVID PCR test before surgery and there were no restrictions on the laparoscopic appendectomy surgery when the PCR result was negative. For this additional reason, it seems to be an advantage that minimally invasive surgery can be performed after a patient has a negative COVID-19 PCR test before surgery.

Fortunately, there were no COVID-19-infected patients among the “COVID-19” group in our hospital. However, if the initial COVID-19 PCR test is positive, it is recommended to perform a second COVID-19 PCR test due to the possibility of false positives. If a COVID-19 result was positive in a second test and the COVID-19-infected patient needed emergency surgery, the surgery would have to be performed in an isolated negative pressure operating room with full-body respiratory protection for the medical staff. One case report has been published of an appendectomy that was performed by medical staff who wore full-body respiratory protection in South Korea [38].

This study had several limitations. Due to the retrospective nature of the study, selection bias cannot be excluded. However, our study had large numbers of patients in the “COVID-19” group compared with previous studies. Currently, in January 2022 and still in the midst of the COVID-19 global pandemic, it is very difficult to conduct a prospective randomized study of COVID-19 status in appendicitis patients. In the same context, the appendectomy surgeries in our study were performed by several general surgeons, including residents (for educational purposes), and the surgical and postoperative treatment protocols were not identical among the patients. The next study should be conducted by adjusting the surgeon factor after including cases from a limited number of surgeons.

## 5. Conclusions

This study is the first large-volume study on whether the COVID-19 pandemic affected the emergency surgery protocols and changed the postoperative clinical courses of patients in one tertiary hospital in South Korea. During the COVID-19 pandemic, a 3 h in-hospital delay for emergency appendectomies due to waiting times for COVID-19 PCR tests did not significantly change the postoperative clinical outcomes of appendicitis patients. Under the COVID-19 situation, we suggest that confirming preoperative COVID-19 PCR tests before proceeding with appendectomies may be safe for both patients and medical staff.

## Figures and Tables

**Table 1 medicina-58-00783-t001:** Clinical characteristics of the appendicitis patients.

Variables	Pre-COVID-19 (*n* = 444)	COVID-19 (*n* = 393)	*p*-Value
Age (years, mean ± sd)	39.13 ± 20.68	38.99 ± 20.57	0.925
Gender			
Male	230 (51.8%)	208 (52.9%)	0.793
Female	214 (48.2%)	185 (47.1%)	
ASA score	2.09 ± 0.62	1.96 ± 0.48	0.001
I	67 (15.0%)	53 (13.5%)	<0.001
II	270 (60.5%)	303 (77.1%)	
III	109 (24.4%)	37 (9.4%)	
BMI (kg/m^2^)	23.39 ± 4.15	23.52 ± 4.28	0.676
COVID-19 PCR result (positive patient/total patients)	0/0	0/393	NA
Operation time (minutes, mean ± sd)	70.62 ± 30.96	58.37 ± 30.84	<0.001
Time spent in ER before surgery (minutes, mean ± sd)	519.11 ± 486.57	705.27 ± 512.59	<0.001
Hospital stay (days, mean ± sd)	4.31 ± 3.30	4.03 ± 2.51	0.176

ASA: American Society of Anesthesiology; ASA I: A normal healthy patient; ASA II: A patient with mild systemic disease; ASA III: A patient with severe systemic disease; BMI: Body mass index, kg/m^2^; ER: Emergency room; OR: Operating room; PCR: Polymerase Chain Reaction; NA: Not applicable.

**Table 2 medicina-58-00783-t002:** Pathological and surgical characteristics of the appendicitis patients.

Variables	Pre-COVID-19 (*n* = 444)	COVID-19 (*n* = 393)	*p*-Value
**Severity of appendicitis**			0.154
Uncomplicated appendicitis (inflammatory, suppurative)	356 (81.1%)	299 (77.1%)	
Complicated appendicitis (gangrenous, perforated)	83 (18.9%)	89 (22.9%)	
**Pathological diagnosis**			NA
Normal to mild inflammation	9 (2.0%)	6 (1.5%)	
Suppurative appendicitis	347 (77.8%)	293 (74.6%)	
Gangrenous appendicitis	76 (17.0%)	51 (13.0%)	
Perforated appendicitis	7 (1.6%)	38 (9.7%)	
Other	7 (1.5%)	5 (1.3%)	
**Extent of surgery**			0.687
**Appendectomy only**	426 (95.5%)	373 (94.9%)	
Open appendectomy	15 (3.4%)	9 (2.3%)	
Laparoscopic appendectomy	411 (92.1%)	364 (92.6%)	
**Extended surgery**	20 (4.5%)	20 (5.1%)	
Partial cecectomy	12 (2.7%)	12 (3.1%)	
Ileocecectomy	2 (0.5%)	6 (1.5%)	
Right hemicolectomy	3 (0.7%)	1 (0.3%)	
Other	3 (0.7%)	1 (0.3%)	
**Surgical approach**			0.799
**Open surgery**	35 (7.8%)	29 (7.4%)	
**Laparoscopic surgery**	411 (92.2%)	364 (92.6%)	
Multiport laparoscopic appendectomy	285 (63.9%)	185 (47.1%)	
Single-port laparoscopic appendectomy	126 (28.3%)	179 (45.5%)	
**Drain placement**	166/444 (37.4%)	105/393 (26.7%)	<0.001
**Readmission within 30 days**	4/444 (0.9%)	13/393 (3.3%)	0.014
**Readmission cause**			NA
Ileus	2	1	
Uncontrolled pain	1	5	
Fluid collection in the abdominal cavity	0	5	
Incisional problem	1	2	

NA: Not applicable.

**Table 3 medicina-58-00783-t003:** Laboratory test results of the patients.

Variables	Pre-COVID-19 (*n* = 444)	COVID-19 (*n* = 393)	*p*-Value
**Preoperative period**			
WBC	12,646.78 ± 4592.34	12,663.03 ± 4043.20	0.957
Hb	13.96 ± 1.78	13.92 ± 1.68	0.728
ANC	10,270.11 ± 4519.71	10,235.77 ± 3922.55	0.906
CRP	4.89 ± 7.14	4.33 ± 5.60	0.205
**Postoperative second day**			
WBC	7066.41 ± 2506.62	7451.68 ± 2938.57	0.042
Hb	12.32 ± 4.74	12.79 ± 4.27	0.133
ANC	4907.66 ± 2346.52	5278.78 ± 2864.41	0.04
CRP	7.98 ± 7.36	8.29 ± 7.79	0.55

WBC: White blood cell count (/µL); Hb: Hemoglobin (g/dL); ANC: Absolute neutrophil count (/µL); CRP: C-reactive protein (mg/L).

## Data Availability

Data sharing not applicable.

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
