# Peer review of "Clinical Experience of Emergency Appendectomy under the COVID-19 Pandemic in a Single Institution in South Korea"

_medicina, 2022, doi:10.3390/medicina58060783_

Round 1
Reviewer 1 Report
The authors evaluated how the COVID-19 pandemic has changed the treatment of patients with acute appendicitis in a single institution in South Korea.
The study is well designed and of importance. However, several issues need to be improved before any favorable decision should be made.
My concerns are as follows:
- Title – Please remove full-stop at the end of the title.
- Introduction – The authors stated that to reduce the complications associated with appendicitis, the diagnosis and surgical treatment of appendicitis should be performed quickly and without delay but they did not specify which diagnostic modalities should be used. I would suggest to add following: ‘’Several diagnostic modalities are available but most frequently laboratory inflammatory markers, different scoring systems and abdominal ultrasonography have been used (REFERENCES: ‘’validity of Appendicitis Inflammatory Response Score in Distinguishing Perforated from Non-Perforated Appendicitis in Children. Children (Basel). 202;8(4):309.’’ AND ‘’Comparison of Open and Laparoscopic Appendectomy in Children: A 5-year Single Center Experience. Indian Pediatr. 2019;56(4):299-303’’).
- Introduction. The authors stated that delay in hospital admissions of the patients with appendicitis have led to increased complication rates. It is important to emphasize that delay in hospital admissions was found for different life-threatening or emergency medical conditions such as acute myocardial infarction (REF: Fox DK, Waken RJ, Johnson DY, Hammond G, Yu J, Fanous E, Maddox TM, Joynt Maddox KE. Impact of the COVID-19 Pandemic on Patients Without COVID-19 With Acute Myocardial Infarction and Heart Failure. J Am Heart Assoc. 2022, e022625. doi: 10.1161/JAHA.121.022625.) or testicular torsion (REF: Pogorelić Z, Milanović K, Veršić AB, Pasini M, Divković D, Pavlović O, Lučev J, Žufić V. Is there an increased incidence of orchiectomy in pediatric patients with acute testicular torsion during COVID-19 pandemic?-A retrospective multicenter study. J Pediatr Urol. 2021;17(4):479.e1-479.e6. doi: 10.1016/j.jpurol.2021.04.017.). Please add this statement and references in the introduction.
- Titles of investigated groups – I would suggest change in titles of investigated groups as follow: ‘’Before COVID-19’’ should be ‘’Pre-COVID-19’’ and ‘’After COVID-19” should be ‘’COVID-19’’ group because patients were treated during COVID-19 pandemic.
- Methodology – Please indicate primary and secondary outcomes of the study in a separate paragraph.
- Methodology – Please indicate inclusion and exclusion criteria for the study.
- Statistical analysis – Please indicate which statistical test was used to test normality of distribution of the data.
- Please replace ethical approval for this study into the first chapter under methodology and add date of approval.
- Did the authors use non-operative treatment for acute appendicitis during the COVID-19 pandemic? Recent meta-analysis which investigated this topic suggested that there was an increased proportion of non-operative management of acute appendicitis during COVID-19 pandemic. Please comment on this. Incidence of Complicated Appendicitis during the COVID-19 Pandemic versus the Pre-Pandemic Period: A Systematic Review and Meta-Analysis of 2782 Pediatric Appendectomies. Diagnostics (Basel). 2022;12(1):127. doi: 10.3390/diagnostics12010127.
- Also, it is unclear which type of appendectomy was performed (open or laparoscopic)? This should be clearly stated under the methodology with description of surgical procedure or adequate reference.
- Please can you comment why the mean hospital stay was 4 days, especially during the COVID-19 pandemic? In most of the centers worldwide the mean duration of hospital stay after appendectomy is 1-2 days. Please comment in the discussion.
- I am surprised that about 5% of the patients from both groups needed extended surgery such as hemicolectomy. Please can you comment on this in your discussion?
- Expression ‘’ileocectomy’’ is inappropriate. Did the authors mean ileocecectomy?
- Table 2 – There is no need to state YES and NO categories – Provide only the number of patients who received investigated intervention. It is understood that if for example 2% of patients had investigated intervention that the other 98% did not. Please revise.
- The authors presented their Readmission cases, which is excellent. They should compare their findings with a recently published study regarding 30-day readmissions: Jukic M et al. Incidence and causes of 30-day readmission rate from discharge as an indicator of quality care in pediatric surgery. Acta Chir Belg. 2021; doi: 10.1080/00015458.2021.1927657.
- For each investigated parameter in discussion the authors should compare their findings with previously published studies!
- The most important systematic review and meta-analysis summarizing all published studies regarding this topic has been published. This should be mentioned in discussion and included in references.REFERENCE: Incidence of Complicated Appendicitis during the COVID-19 Pandemic versus the Pre-Pandemic Period: A Systematic Review and Meta-Analysis of 2782 Pediatric Appendectomies. Diagnostics (Basel). 2022;12(1):127. doi: 10.3390/diagnostics12010127.
- Limitations of the study – This statement is contradictory: ‘’ This study collected patients prospectively, but this study was a retrospective analysis’’. Do the authors mean one arm of the study was prospective? Please revise.
- Reference list should be updated. There are numerous studies regarding investigated topics which should be included and commented on in discussion.
- Quality of English should be improved.Manuscript should be edited for the language from a native English speaker or professional language editing service to improve the grammar and readability.
Author Response
The authors evaluated how the COVID-19 pandemic has changed the treatment of patients with acute appendicitis in a single institution in South Korea.
The study is well designed and of importance. However, several issues need to be improved before any favorable decision should be made.
My concerns are as follows:
- Title – Please remove full-stop at the end of the title.
Sincerely for your comments.
We removed full-stop as your recommend.
- Introduction – The authors stated that to reduce the complications associated with appendicitis, the diagnosis and surgical treatment of appendicitis should be performed quickly and without delay but they did not specify which diagnostic modalities should be used. I would suggest to add following: ‘’Several diagnostic modalities are available but most frequently laboratory inflammatory markers, different scoring systems and abdominal ultrasonography have been used (REFERENCES: ‘’validity of Appendicitis Inflammatory Response Score in Distinguishing Perforated from Non-Perforated Appendicitis in Children. Children (Basel). 202;8(4):309.’’ AND ‘’Comparison of Open and Laparoscopic Appendectomy in Children: A 5-year Single Center Experience. Indian Pediatr. 2019;56(4):299-303’’).
Sincerely for your comments.
We followed as your recommends and put the following in the introduction and the reference you recommended.
Several diagnostic modalities are available but most frequently laboratory inflammatory markers, different scoring systems, computered tomography(CT) scan and abdominal ultrasonography have been used.
Added content is indicated in red text.
- Introduction. The authors stated that delay in hospital admissions of the patients with appendicitis have led to increased complication rates. It is important to emphasize that delay in hospital admissions was found for different life-threatening or emergency medical conditions such as acute myocardial infarction (REF: Fox DK, Waken RJ, Johnson DY, Hammond G, Yu J, Fanous E, Maddox TM, Joynt Maddox KE. Impact of the COVID-19 Pandemic on Patients Without COVID-19 With Acute Myocardial Infarction and Heart Failure. J Am Heart Assoc. 2022, e022625. doi: 10.1161/JAHA.121.022625.) or testicular torsion (REF: Pogorelić Z, Milanović K, Veršić AB, Pasini M, Divković D, Pavlović O, Lučev J, Žufić V. Is there an increased incidence of orchiectomy in pediatric patients with acute testicular torsion during COVID-19 pandemic?-A retrospective multicenter study. J Pediatr Urol. 2021;17(4):479.e1-479.e6. doi: 10.1016/j.jpurol.2021.04.017.). Please add this statement and references in the introduction.
We really appreciate your comments.
We put the following contents in the introduction and the reference you recommended
In other emergency diseases, myocardial infarction and testicular torsion, there has been some reports that the incidence of disease complications increased during the COVID-19 pandemic.
Added content is indicated in red text.
- Titles of investigated groups – I would suggest change in titles of investigated groups as follow: ‘’Before COVID-19’’ should be ‘’Pre-COVID-19’’ and ‘’After COVID-19” should be ‘’COVID-19’’ group because patients were treated during COVID-19 pandemic.
Your comments are consistent with other reviewer.
We have revised "Before COVID-19" to "Pre COVID-19" and "After COVID-19" to "COVID-19" based on your comments.
We've also shared your comments with other reviewers.
Thank you for your kindness.
- Methodology – Please indicate primary and secondary outcomes of the study in a separate paragraph.
Sincerely for your comments.
We divided the paragraph of methodology into two paragraphs according to the primary and secondary outcomes as follows according to your opinion.
We collected the clinical data, including the patients’ general characteristics (age, sex and body mass index (BMI, kg/m2), the American Society of Anesthesiology (ASA) score); the time spent in the emergency room (ER) before the surgery; the operation-related variables (operation time, surgical approach, surgical extent and drain placement)); and the laboratory values (WBC (white blood cell), Hb (hemoglobin), ANC (absolute neutrophil count) and CRP (C-reactive protein)) on the day before the surgery and the second day after the surgery.
We also collected the pathologic results of the appendicitis patients after surgery and the factors that were associated with postoperative hospital readmissions within 30 days after surgery.
Revised content is indicated in red text.
- Methodology – Please indicate inclusion and exclusion criteria for the study.
We really appreciate your comments.
We included only patients diagnosed with acute appendicitis who underwent emergent surgery and excluded patients who underwent delayed appendectomy or other medical management.
We'll insert the following contents into the method.
Patients who underwent delayed appendectomy and medical management for various reasons were excluded.
Added content is indicated in red text.
- Statistical analysis – Please indicate which statistical test was used to test normality of distribution of the data.
We respond to your comments.
We used student T-test and X square method to compare both group in this study.
If sample size is over 30, normality can be assumed usually without testing, in student T-test. If sample size is between 10 and 30, Kolmogorov-Smirnov test or Shapiro-Wilk test is used to test normality of distribution of the data.
This study is a double arm study with about 400 people per arm. Because the volume is sufficient, it is thought that special statistical analysis for normality distribution is not necessary in X square .
Please understand this point.
Thank you for your kindness.
- Please replace ethical approval for this study into the first chapter under methodology and add date of approval.
Thank you for your comments.
According to your opinion, the ethics part has been moved to the front of the methodology and the approval date has been added.
Revised content is indicated in red text.
- Did the authors use non-operative treatment for acute appendicitis during the COVID-19 pandemic? Recent meta-analysis which investigated this topic suggested that there was an increased proportion of non-operative management of acute appendicitis during COVID-19 pandemic. Please comment on this. Incidence of Complicated Appendicitis during the COVID-19 Pandemic versus the Pre-Pandemic Period: A Systematic Review and Meta-Analysis of 2782 Pediatric Appendectomies. Diagnostics (Basel). 2022;12(1):127. doi: 10.3390/diagnostics12010127.
We really appreciate your comments.
In South Korea, the basic treatment for acute appendicitis is surgery. The main reason is that the surgical cost is very low and the surgical results are very good. Although the treatment may be different for each situation, the basic treatment principle is surgical appendectomy, and in our hospital, surgical appendectomy is the treatment principle.
The topic of this paper is the impact of the COVID-19 pandemic on surgical appendectomy. But, the thesis you recommended is a very good, and I will attach it to the introduction part with other reference.
There were several reports that complicated acute appendicitis and non-surgical treatment has increased compared to pre COVID-19 pandemic.
Added content is indicated in red text.
And the meta-analysis paper you recommended is a very good paper, so I will mention it separately in the discussion and put it as a reference.
Thank you for your kindness
- Also, it is unclear which type of appendectomy was performed (open or laparoscopic)? This should be clearly stated under the methodology with description of surgical procedure or adequate reference.
Sincerely for your comments.
As your opinion, we inserted appendectomy type results(open or laparoscopic) in table 2
We inserted the following into the method
Surgical approach is defined which method is used to perform surgery.
Open appendectomy was performed using the open method, and laparoscopic appendectomy was performed using the laparoscopic method.
Added content is indicated in red text.
- Please can you comment why the mean hospital stay was 4 days, especially during the COVID-19 pandemic? In most of the centers worldwide the mean duration of hospital stay after appendectomy is 1-2 days. Please comment in the discussion.
We really appreciate your comments.
In our hospital, the total hospital stay is calculated from the day patients are admitted to the emergency room to the day patients are discharged after surgery.
If you consider the hospital days after surgery, it will take about 2-3 days.
Appendectomy is often performed by a general surgeon as well as a gastrointestinal part specialized surgeon. (even if breast or thyroid specialized surgeon)
It is also often implemented for resident education.
(Out hospital is one of educational hospital in South Korea.)
Because it is performed by several different surgeons, post-operative management is also performed in a variety of ways.
Please understand the points.
- I am surprised that about 5% of the patients from both groups needed extended surgery such as hemicolectomy. Please can you comment on this in your discussion?
Thank you for your good comments.
Extended surgery in our sutdy included partial cecectomy, ileocectomy and right hemicolectomy.
We will insert the following into the discussion based on your comments.
There was no statistically significant difference the rate of extended surgery beyond simple appendectomy at about 5% between the “Pre COVID-19” and the ”COVID-19” groups.
Added content is indicated in red text.
- Expression ‘’ileocectomy’’ is inappropriate. Did the authors mean ileocecectomy?
We really appreciate your comments.
It`s our mistake. We will revise it.
Thank you for your kindness
- Table 2 – There is no need to state YES and NO categories – Provide only the number of patients who received investigated intervention. It is understood that if for example 2% of patients had investigated intervention that the other 98% did not. Please revise.
Thank you for your good comments.
Table 2 has been revised as your comments.
Revised content is indicated in red text.
Thank you for your kindness
- The authors presented their Readmission cases, which is excellent. They should compare their findings with a recently published study regarding 30-day readmissions: Jukic M et al. Incidence and causes of 30-day readmission rate from discharge as an indicator of quality care in pediatric surgery. Acta Chir Belg. 2021; doi: 10.1080/00015458.2021.1927657.
Thank you so much for the good reference recommendation.
We will add the following content to the discussion and insert the reference you recommended.
This readmission rate is relative higher than that reported in other study.
Added content is indicated in red text.
- For each investigated parameter in discussion the authors should compare their findings with previously published studies!
We really appreciate your comments.
We started preparing for this study in May 2021 and completed the paper in November 2021.
When we write our paper, there were not published many studies yet and their quality were not good.
As you can see in your meta-analysis, the number of patients included in our study is very large.
When we were preparing, we thought that this study is valuable because it had relatively large number of patients in a short period within single center.
Please understand this point.
Thank you for your kindness
- The most important systematic review and meta-analysis summarizing all published studies regarding this topic has been published. This should be mentioned in discussion and included in references.REFERENCE: Incidence of Complicated Appendicitis during the COVID-19 Pandemic versus the Pre-Pandemic Period: A Systematic Review and Meta-Analysis of 2782 Pediatric Appendectomies.
Diagnostics (Basel). 2022;12(1):127. doi: 10.3390/diagnostics12010127.
We really appreciate your comments.
The meta-analysis study you recommended is a very good study.
However, I think that the impact of the COVID-19 pandemic on medical care may be different for each country and each situation.
The study you recommended has a different conclusion from our study, but I think that it is a very valuable study.
As your recommend, we have inserted separately the following content into the discussion and the reference you recommended.
There was a large volume meta-analysis that COVID pandemic increased complicated appendicitis and non-operative appendicitis management.
Added content is indicated in red text.
Thank you for your kindness
- Limitations of the study – This statement is contradictory: ‘’ This study collected patients prospectively, but this study was a retrospective analysis’’. Do the authors mean one arm of the study was prospective? Please revise.
Sincerely for your comments.
We revised as your recommend
This study was a retrospective analysis, problems such as selection bias caused by retrospective nature cannot be excluded.
Revised content is indicated in red text.
Thank you for your kindness
- Reference list should be updated. There are numerous studies regarding investigated topics which should be included and commented on in discussion.
As you recommended, we have added some studies as references.
We really appreciate your comments.
We started preparing for this study in May 2021 and completed the paper in November 2021.
When we write our paper, there were not published many studies yet and their quality were not good.
As you can see in your meta-analysis, the number of patients included in our study is very large.
When we were preparing, we thought that this study is valuable because it had relatively large number of patients in a short period within single center.
Please understand this point.
Nowadays, Korea is preparing to end the COVID pandemic and transition to endemic status.
As the next step study, we would like to review the treatment outcomes of acute appendicitis during the entire COVID pandemic.
In the next study, we will list up a new reference, as you recommended.
Thank you for your kindness.
- Quality of English should be improved. Manuscript should be edited for the language from a native English speaker or professional language editing service to improve the grammar and readability.
First of all, I would like to apologize for my lack of English skills.
We submitted to MEDICINA journal after verification AJE (American Journal Expert) for English proofreading.
(AJE verified date: February 7, 2022 ; Verification code : E714-6DD9-DFB5-1CD4-2F1P)
I will try harder to submit better papers in the future.
I sincerely appreciate you.

Reviewer 2 Report
The authors have presented evidence that a moderate delay in ER during the pandemic does not compromise the surgical condition.
There are a few points about the language which need to be looked into:
- Has the duration of symptom before operation actually shortened during the pandemic, because the patients present earlier to hospital?
- Suggest replacing "after Covid pandemic" to after the start of Covid pandemic, as we are still suffering from the pandemic currently
- suggest replacing "Surgical extent" by extent of surgery, or extent of surgical procedure
- how would your finding change your current surgical management of suspected appendicitis?
Author Response
The authors have presented evidence that a moderate delay in ER during the pandemic does not compromise the surgical condition.
There are a few points about the language which need to be looked into:
- Has the duration of symptom before operation actually shortened during the pandemic, because the patients present earlier to hospital?
Sincerely for your comments.
In our study, the time from the o nset of symptoms to the time of admission was not known.
So I can't answer your question.
Thank you for your kindness.
- Suggest replacing "after Covid pandemic" to after the start of Covid pandemic, as we are still suffering from the pandemic currently
Your comments are consistent with other reviewer.
According to your recommend and other reviewer recommend, We have revised "Before COVID-19" to "Pre COVID-19" and "After COVID-19" to "COVID-19"
Thank you for your good comments.
- suggest replacing "Surgical extent" by extent of surgery, or extent of surgical procedure
We really appreciate your comments.
As your recommend, we will replace "surgical extent" with "extent of surgery".
Thank you for your good comments.
- how would your finding change your current surgical management of suspected appendicitis?
Sincerely for your comments.
Thank you very much for your questions. As claimed in this study, it is thought that there will be no problems in south korea if you do it as you have done so far.
It is the subject of this study that some time delay in acute appendicitis tratement is acceptable for medical staff to perform PCR tests on all patients in order to prevent the spread of COVID-19 infection and nosocomial transmission.
I sincerely appreciate you.
Round 2
Reviewer 1 Report
The authors significantly improved the manuscript. However several important points have not been addressed adequately:
1) Outcomes of the study – The authors should clearly state what were primary and secondary outcomes of the study. E.g. Primary outcome of the study was incidence of complicated appendicitis in the investigated group and secondary outcomes were length of hospital stay, rate of complications, number of readmissions…
2) The authors were asked regarding the approach (open vs. laparoscopic). They stated: ‘’Open appendectomy was performed using the open method, and laparoscopic appendectomy was performed using the laparoscopic method.’’. This answer is premature. They should state. Both, open and laparoscopic appendectomy were used, based on the operating surgeon decision. They should describe each procedure or provide adequate reference where these procedures were explained in details (e.g. ref. No 7 from reference list)
3) Length of hospital stay – The authors should explain in methodology that length of hospital stay was calculated from admission at the emergency department (not after surgery) because readers should get the wrong impression that post-surgical length of hospital stay was 4 days.
4) The authors were asked to explain why about 5% of the patients from both groups needed extended surgery such as hemicolectomy. They just confirmed that statement, but I think they need to explain indications for extended surgery. In my 15-years of surgery as surgeon I performed only once or twice extended surgery due to acute appendicitis, so this number of 5% is quite uncommon. Please can you also comment on this in your discussion?
5) Discussion should be improved. The authors should compare their main outcomes such as number of complicated appendicitis, length of hospital stay, number of complications / reoperations/ readmission with other published studies. Discussion in this form is not so good.
6) English should be improved, especially in newly added text.
Author Response
The authors significantly improved the manuscript. However several important points have not been addressed adequately:
1) Outcomes of the study – The authors should clearly state what were primary and secondary outcomes of the study. E.g. Primary outcome of the study was incidence of complicated appendicitis in the investigated group and secondary outcomes were length of hospital stay, rate of complications, number of readmissions
Sincere thanks for your accurate point
We followed as your recommends and put the following in the method as you recommended.
The primary outcome in this study is the rate of complicated appendicitis and the secondary outcomes are the length of hospital stay, the rate of extended surgery, and the readmission rate.
Added content is indicated in blue text.
2) The authors were asked regarding the approach (open vs. laparoscopic). They stated: ‘’Open appendectomy was performed using the open method, and laparoscopic appendectomy was performed using the laparoscopic method.’’. This answer is premature. They should state. Both, open and laparoscopic appendectomy were used, based on the operating surgeon decision. They should describe each procedure or provide adequate reference where these procedures were explained in details (e.g. ref. No 7 from reference list)
We really appreciate your comments.
We followed as your recommends and put the following in the method
Surgical method
The surgical approach was defined on which method was used to the perform surgery, open or laparoscopic. The choice of surgical approach was decided by the individual operating surgeon. All surgeries were performed under emergency.
Open appendectomy: A 5-7cm modified Mcburney or Rocky-Davis incision was made depending on surgeon`s decision. After opening the abdominal wall, the appendix was identified and the periappendiceal tissue dissection was done. The appendiceal vessel was ligated with absorbable suture. The appendix base was ligated using a double tie method, transected and removed. The exposed mucosa was cauterized with an electrosurgical device such as Bovie. Stump inversion was done by placing a purse-string suture.
Laparoscopic appendectomy: The infra umbilical port was inserted initially for the laparoscope. After CO2 was insufflated at a pressure of 12mmHg, additional 2 ports were inserted (5mm 2 sites or 5mm, 12mm depend on the surgeon`s preference). Under the laparoscope, the appendix was identified and the periappendiceal tissue dissection was done. The mesoappendix and appendiceal vessel were dissected using a ultrasonic shear energy device (Harmonic, Ethicon Endo-surgery, Cincinnati, OH, USA). The appendiceal base was tied using endoloops (Vycril Endoloop-0, Ethicon Endo-surgery, Cincinnati, Ohio, USA) or polymetric clips (Hem-o-lock, Teleflex, Morrisville, North Carolina, USA). The appendiceal mucosal tip was cauterized with an electrical device such as Bovie.
Single port laparoscopic appendectomy: A single port (Gloveport, Nelis, Bucheon, Gyeonggi-do, Korea) was used for laparoscopy instead of the traditional 3 ports. 2-3cm size transumbicial incision was made and the single port was inserted for laparoscope. Other surgical methods are consistent with the conventional laparoscopic appendectomy.
Added content is indicated in blue text.
I really appreciate your advice
Thanks for the good comments.
3) Length of hospital stay – The authors should explain in methodology that length of hospital stay was calculated from admission at the emergency department (not after surgery) because readers should get the wrong impression that post-surgical length of hospital stay was 4 days.
Thank you for your comments.
We have inserted the following into the method according to your comments.
"Hospital stay" was defined as from the day of admission to the emergency room to the day of discharge after surgery.
Sincere thanks for your comment
4) The authors were asked to explain why about 5% of the patients from both groups needed extended surgery such as hemicolectomy. They just confirmed that statement, but I think they need to explain indications for extended surgery. In my 15-years of surgery as surgeon I performed only once or twice extended surgery due to acute appendicitis, so this number of 5% is quite uncommon. Please can you also comment on this in your discussion?
Thank you for your comment
Extended surgery in our study included partial cecectomy, ileocecectomy and right hemicolectomy.
Almost of them were partial cecectomy patients due to cecum base inflammation.
Ileocecectomy cases were about 0.5-1.5%.
The Right hemicolectomy cases that you mentioned were about 0.3-0.7% of total appendicitis case.
In a study published at another hospital in South Korea around the same time, the incidence of ileocecectomy was reported to be 0.6-0.9%.
Rather, the rate of cecectomy was reported as high as 14.5–14.9%.
Please understand kindly this point.
Attaching the study we mentioned.(reference No. 31)
Kim, C.W.; Lee, S.H. Impact of COVID-19 on the care of acute appendicitis: a single-center experience in Korea. Annals of surgical treatment and research 2021, 101, 240-246, doi:10.4174/astr.2021.101.4.240.
5) Discussion should be improved. The authors should compare their main outcomes such as number of complicated appendicitis, length of hospital stay, number of complications / reoperations/ readmission with other published studies. Discussion in this form is not so good.
We really appreciate your comments.
We followed as your recommends and put the following and some reference in the discussion.
There was a large volume meta-analysis that during the COVID-19 pandemic the rate of complicated appendicitis and non-operative management of appendicitis increased. [22] And there are also many other reports that the incidence of complicated appendicitis has increased compared to that before the COVID-19 pandemic. [12-14,16,23-30] Their HR (hazard ratios) compared to those of pre COVID-19 pandemic patients were from 1.13 to 3.15(average HR=1.63, CI=1.33-2.01; p<0.001). Factors that may affect the increase in complicated appendicitis include the increase in the duration from symptom onset to the time of admission due to social distancing or isolation and delay in time from admission to surgery because of hospital pandemic protocols. [23] These factors may differ by country, time and etc. There was a study similar to our study conducted in Korea at the similar time. [31] They reported that the rate of complicated appendicitis increased compared to the pre COVID-19 pandemic (82/161(50.9%) versus 201/330(60.9%); p<0.036). However, there were no differences in postoperative length of hospital stay and complication rates between the two groups. The readmission rate within 30 days was lower compared to the pre COVID-19 pandemic (6/161(3.7%) versus 2/330(0.7%); p=0.017).
Added content is indicated in blue text.
Thanks for the good comments.
6) English should be improved, especially in newly added text.
First of all, I would like to apologize for my poor English skills.
One of our authors, Chris Tae Young Chung, helped with the English revision.
He is a US citizen and one of the most fluent English surgeons.
Regarding the points you pointed out, English revision was performed with the help of Dr. Chung.
As you pointed out, my English is not good. I try to work harder to write better papers in next time.
Thank you for your kindness.
